# Automated identification of piglet brain tissue from MRI images using Region-based Convolutional Neural Networks

**Kayla L. Stanke[1], Ryan J. Larsen[1]\*, Laurie Rund[1], Brian J. Leyshon[2], Allison Y. Louie[3], Andrew J. Steelman[1,3,4,5]**

1 Department of Animal Sciences, University of Illinois Urbana-Champaign, Champaign, Illinois, United States of America, 2 Abbott Nutrition, Discovery Research, Columbus, Ohio, United States of America, 3 Division of Nutritional Sciences, University of Illinois Urbana-Champaign, Champaign, Illinois, United States of America, 4 Neuroscience Program, University of Illinois Urbana-Champaign, Champaign, Illinois, United States of America, 5 Carl R. Woese Institute for Genomic Biology, University of Illinois Urbana-Champaign, Champaign, Illinois, United States of America

\* larsen@illinois.edu

**Data Availability Statement:** All relevant data have been uploaded to the Illinois Data Bank: https://doi.org/10.13012/B2IDB-5784165_V1.

## Abstract

Magnetic resonance imaging is an important tool for characterizing volumetric changes of the piglet brain during development. Typically, an early step of an imaging analysis pipeline is brain extraction, or skull stripping. Brain extractions are usually performed manually; however, this approach is time-intensive and can lead to variation between brain extractions when multiple raters are used. Automated brain extractions are important for reducing the time required for analyses and improving the uniformity of the extractions. Here we demonstrate the use of Mask R-CNN, a Region-based Convolutional Neural Network (R-CNN), for automated brain extractions of piglet brains. We validate our approach using Nested Cross-Validation on six sets of training/validation data drawn from 32 pigs. Visual inspection of the extractions shows acceptable accuracy, Dice coefficients are in the range of 0.95–0.97, and Hausdorff Distance values in the range of 4.1–8.3 voxels. These results demonstrate that R-CNNs provide a viable tool for skull stripping of piglet brains.

## Introduction

Piglets are an important translational model for measuring the effect of nutrition on brain development. Not only do piglet brain stages of development correlate with human infant development, but their nutritional requirements are also comparable [1]. Magnetic resonance imaging (MRI) is an important technique for obtaining non-invasive measurements of brain volumes. An early step in volumetric analysis is the identification or "extraction" of the brain from the surrounding tissue. Manual tracing has been the gold standard for brain extraction and is performed by creating an outline that separates the surrounding skull, muscles, tissues, and fat from the brain [1–6]. However, the method is not ideal when working with large data sets because it is time intensive and is subject to inconsistencies between raters/evaluators. Automated brain extraction techniques are needed to overcome these limitations. However,

**Funding:** The study was funded by Abbott Nutrition, www.abbott.com, grant number 00490326 to A.S. The funder aided with study design, data analysis, decision to publish, and preparation of the manuscript.

**Competing interests:** The authors have declared that no competing interests exist.

reports of automated extractions of pig brains are limited. A graph theory approach that makes use of prior information of anatomical structures has been used to perform automated extraction of piglet brains [7]. However, deep learning technologies offer the possibility of automating the training of anatomical structures to enable brain extractions [8–10]. A U-Net model trained on humans, non-human primates, and 3 pig scans has been shown to successfully perform brain extractions on 2 pig scans [11]. Also, a patch-based 3D U-Net trained on piglets has been used for successful brain extractions of piglets of multiple ages, via transfer learning [12]. An alternative segmentation tool is Mask R-CNN, a Region-based Convolutional Neural Network (R-CNN). This tool has been used to create a mouse brain atlas that is generalizable across developmental ages and imaging modalities [10]. These results suggest that Mask R-CNN may be effective for piglet brain extraction. Here we demonstrate the use of Mask R-CNN for automated brain extractions of piglet brains.

## Methods

### Animals and care practices

All animal care and handling procedures were approved by the University of Illinois Institutional Animal Care and Use Committee (Protocol #18256) and were in accordance with federal guidelines. Male and female average-for-gestational age, Yorkshire crossbred, full-term, naturally-delivered piglets were obtained from University of Illinois Swine Farm at postnatal day 2 to allow for colostrum consumption. All piglets remained intact but did undergo routine processing on the farm including iron dextran (Henry Schein Animal Health, Dublin, OH, USA) and antibiotic injection (EXCEDE®, Zoetis, Parsippany, NJ 07054, USA) per routine farm practice and according to label. Four groups of piglets were placed individually into a caging system under standard conditions as described in a previous publication [13] and randomly assigned to five diet treatment groups. The control group of piglets remained with the sow until day 28 of age with free access to suckle and were weighed daily. The current study does not distinguish between diet groups.

Sow-raised piglets were handled daily, as were the artificially raised animals to diminish differences between the two types of rearing. After arrival at the Edward R. Madigan Laboratory (ERML) Animal Facility, experimental piglets received one dose of antibiotic: the first of two cohorts received Spectraguard™ (Bimeda, Inc, Oakbrook Terrace, IL 60181) on postnatal day 2 and the second cohort received Baytril® (Bayer healthcare LLC Shawnee Mission, KS 66201) on postnatal day 4. Additional doses of antibiotic were administered during the experiment only under the direction of the staff veterinarian. Animals were individually housed in racks of metabolism cages specifically designed to artificially rear neonatal piglets under constant 12-h light/dark cycles. Piglet housing at ERML was as follows: space allowance for individual piglets was 30" deep, 23" wide, and 18.5" high, providing 4.8 square feet of floor space per piglet. Each piglet was supplied with a heating pad, toy, and blanket. Room temperatures were kept at 85–95˚F using space heaters. Animals were also offered water or BlueLite® electrolyte solution ad libitum. Cages and heating pads were disinfected, and toys and blankets replaced daily. Animal care protocols were in accordance with National Institutes of Health Guidelines for Care and Use of Laboratory Animals and were approved by the University of Illinois Laboratory Animal Care and Use Committee.

### MRI acquisition

MRI data were acquired using 3 T Prisma scanner (Siemens, Erlangen) housed at the Biomedical Imaging Center at the University of Illinois. Pigs were anesthetized using TKX (combination of 2.5 ml of xylazine (100 mg/ml) and 2.5 ml of ketamine (100mg/ml) added to a Telazol

vial and administered at a dosage of 0.02–0.03 ml/kg IM) then maintained on isofluorane (1–3%) throughout the imaging. Animals were scanned in the supine position using a specialized piglet head coil (Rapid Biomed, Rimpar). During scanning, the respiration rate, heart rate and blood oxygen levels were monitored using a LifeWindow LW9x monitor (Digicare, Boynton Beach, FL).

MRI Structural Imaging: Our structural MRI scan consisted of a 3D MPRAGE acquisition (voxel size = 0.6 x 0.6 x 0.6 mm$^3$, FOV = 173 x 173 mm$^2$, 256 slices, GRAPPA—GeneRalized Autocalibrating Partial Parallel Acquisition—acceleration factor R = 2; TR = 2060 ms, TI = 1060 ms, flip angle = 9$^\circ$, for an overall scan time of 5:21 min).

## Manual brain extraction

Manual brain extraction was facilitated by first performing a rigid alignment of $T_1$-weighted images from all piglets to the brain atlas [14]. This was done using SPM12 imaging analysis software. First, we performed a manual rotation to approximately align the $T_1$-weighted images with 28-day piglet template [14], without resampling. We then used the "coregistration" function of SPM12 to further align the $T_1$-weighted images to the template, again without resampling. The resulting alignment was accurate for all piglets, even though it was performed without first performing brain extraction, as shown by a representative image in Fig 1.

An atlas-based brain mask was then resampled into the native space of each piglet, providing initial estimates of the brain masks for each piglet. These initial estimates were then modified to create individualized brain masks using Slicer3D, as shown in Fig 2. Most revisions were done in the sagittal plane, but all three orthogonal views were reviewed and modified for improved precision. All extractions were performed by one rater to minimize variability. No image intensity normalization was performed prior to manual or automated brain extraction.

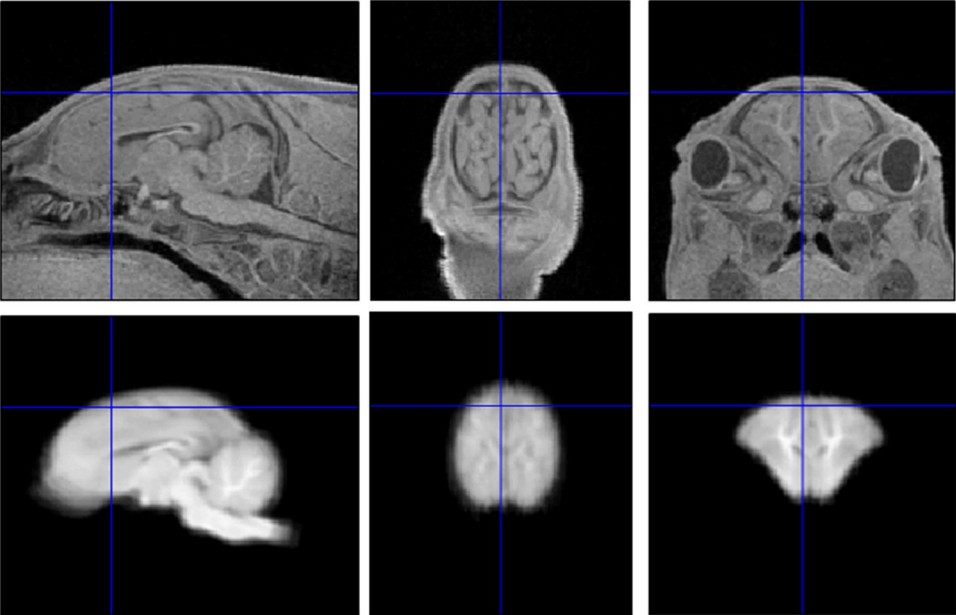

**Fig 1. Results of coregistration of piglet brain to the average brain template.** The top row shows images from a representative piglet brain and bottom row shows the same slices from the average brain template. Blue lines within each image indicate the locations of the perpendicular slices. The consistency of two sets of images confirms a good approximate alignment of the $T_1$-weighted image with the average brain template.

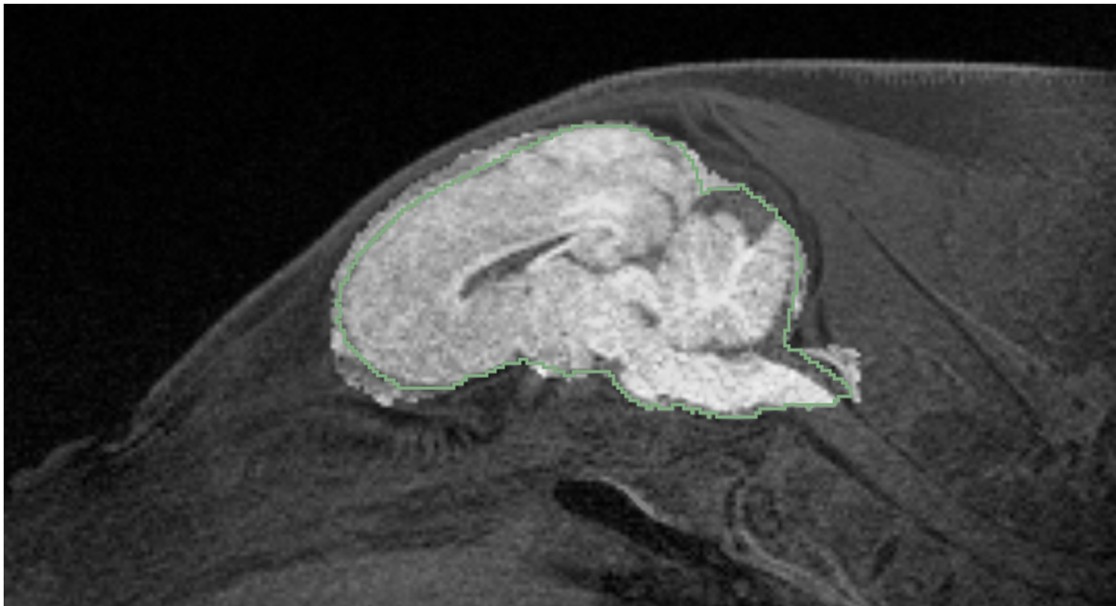

**Fig 2. Demonstration of the creation of manually-defined brain mask via modification of the template brain mask.** A representative $T_1$-weighted image is overlayed with the template brain mask, creating a brighter region. The green outline shows the manual-designated brain mask. The manual brain mask was created by editing the template brain mask for each individual piglet.

## Automated brain extraction

This study uses a deep learning instance segmentation neural network using object detection model Mask R-CNN with Inception Resnet v2 architecture [15], pretrained on the COCO 2017 data set [16]. Our model uses a Tensor Flow 2.4.0 implementation of Mask R-CNN [15], with feature extractor Faster R-CNN [17]. Faster R-CNN utilizes a Region Proposal Network (RPN) to select object proposals from a backbone, which for our study was generated using a combined ResNet101 and Feature Pyramid Network (FPN) [18].

Training was performed using a single NVIDIA GeForce GTX 1070 Ti GPU, with NVIDIA developer driver 465.21, CUDA 11.0, and CUDNN library 8.0.4. The network was trained with a cosine decay learning rate of 0.008 and a momentum optimizer value of 0.9, and a batch size of two images, or slices, per iteration. We performed 200,000 iterations, or 48.8 epochs. The training and segmentation were performed only in 2D sagittal planes. During evaluation, predicted masks were binarized at a confidence parameter threshold of 0.5. The masks created from the 2D slices were then combined into 3D datasets for final cleaning. This involved largest connected component (LCC) filtering to remove several small and spurious masks, typically occurring in slices that did not include brain. Cleaning was done by first using the Matlab function "bwconncomp" to identify all isolated masks, consisting of one or more connected voxels, where connected voxels are defined as those with touching faces. Then we discarded all but the largest mask, or brain mask.

## Validation

Nested Cross-Validation was used to evaluate the performance of the training models. This method has been shown to produce unbiased performance estimates, even in small datasets [19]. We randomly assigned each of the 32 pigs to one of six test groups. Four of the test groups consisted of five pigs, and two of the test groups consisted of six pigs. For each test group, a

training model was generated using the remaining pigs, beginning with the same architecture which was naïve to the test images. Validation of the test groups was performed by comparing the machine-generated masks with the manually generated masks by visual inspection, by computing Dice coefficients, 3D Hausdorff Distance (HD) values, and Pearson correlations between the manual and machine-generated masks. Dice coefficients were calculated using the formula $2|X \cap Y|/(|X|+|Y|)$, where $|...|$ indicates the number of voxels within the mask, $\cap$ indicates the union, and $X$ and Y indicate the manual and machine-generated masks. Dice coefficients were calculated before and after LCC filtering. We calculated 3D Hausdorff Distance (HD) values, using the freely-available "EvaluateSegmentation" command-line tool (https://github.com/Visceral-Project/EvaluateSegmentation) [20]. Because HD values are sensitive to outliers eliminated by LCC filtering, HD values were only calculated after LCC filtering [20].

## Results

Visual inspection of the brain extractions reveals good accuracy of automatic brain extractions (Fig 3). The six models were labelled with letters from A to F. We found that Model D failed to identify the brain within several sagittal slices of one of the test pigs, as shown in Fig 4(A) and 4(B). These slices included an unusually bright region in the subcutaneous fat layer near the superior area of the head (see Fig 4(B)). This bright region was removed by manually outlining it in one of the slices, and then removing the traced voxels from all the slices (see Fig 4(D)). The modified structural images were then re-evaluated using the same model, producing a more accurate brain mask (see Fig 4(C) and 4(D)). This manual correction boosted the Dice coefficient of the pre-LCC filtered images from 0.91 to 0.96. The brain mask from the modified images was used for subsequent validation.

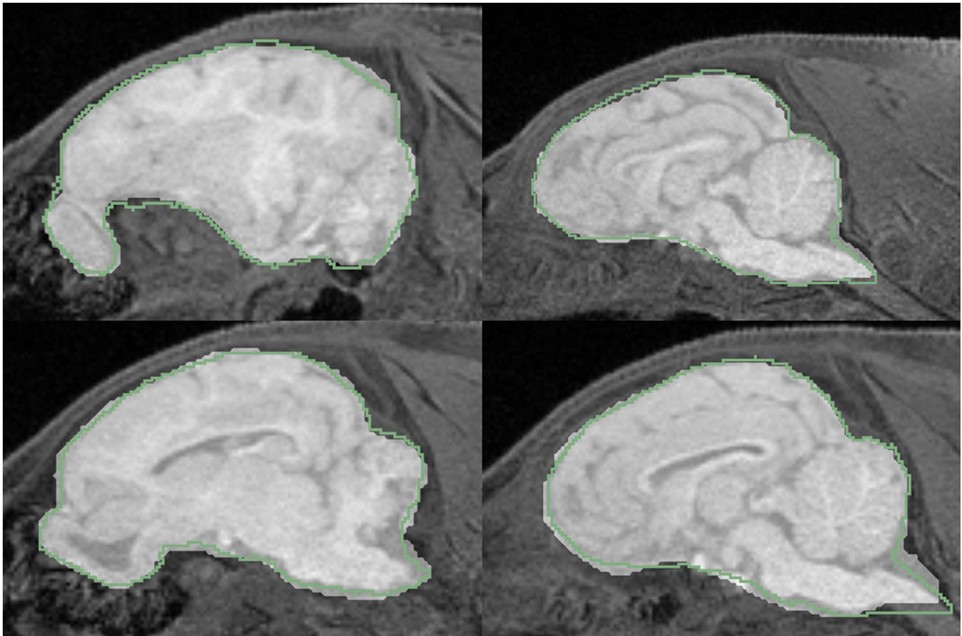

**Fig 3. Sample extractions from two piglets.** The overlay that creates a brighter region indicates machine extractions, and the green outline shows the manual brain masks. The top row shows a piglet tested using Model A, and the bottom row shows a piglet tested using Model B.

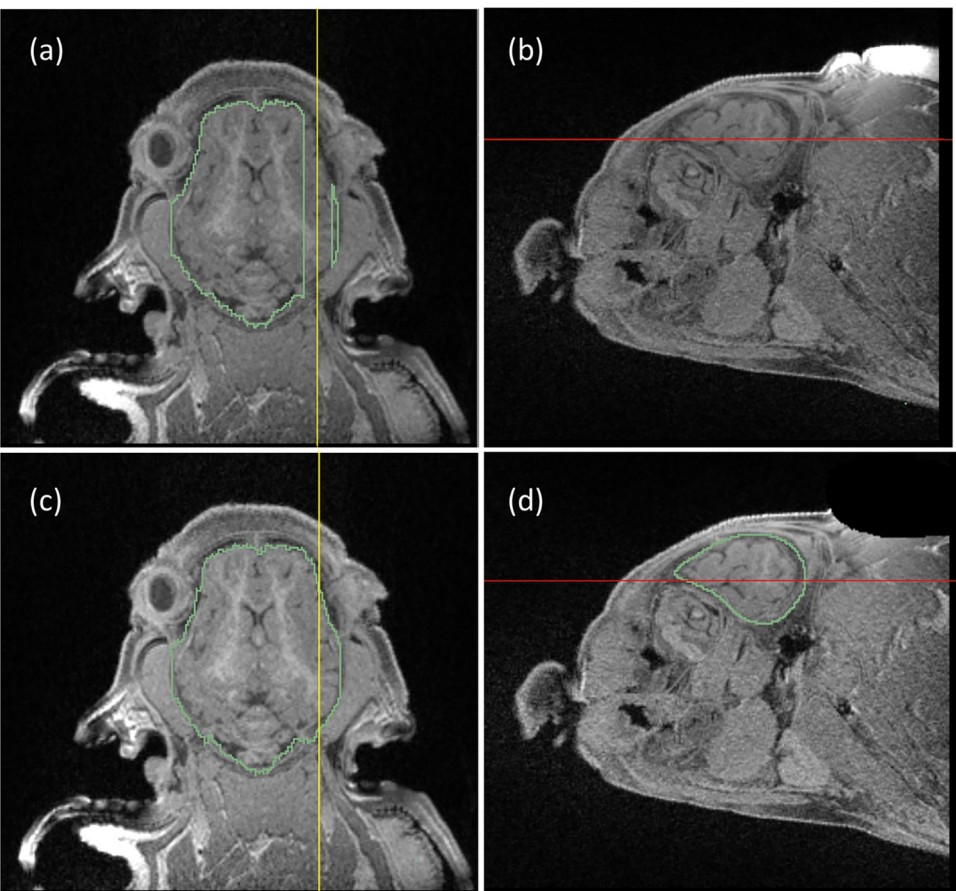

**Fig 4. A segmentation error generated by Model D that was subsequently corrected.** Panels (a) and (b) show the inaccurate mask, indicated by green lines, and panels (c) and (d) show the mask generated by the same model after image modification, with the same views and the same slices. Axial slices are shown in (a) and (c); sagittal slices are shown in (b) and (d). The yellow lines in the axial views, (a) and (c), show the location of the sagittal view and the red lines in the sagittal views, (b) and (d), show the location of the axial views. For the inaccurate mask there are several sagittal slices in which no brain regions were identified, as shown in (a). These slices included a bright region in the fat layer of the superior region of the head, as shown in (b). The bright region was manually traced and removed from the same voxels of all sagittal slices as shown in (d). A re-evaluation of the edited images using the same model, Model D, produced an accurate brain extraction, as shown in (c) and (d).

Visual inspection of the results also revealed that the LCC filter was important for one of the pigs, for which the automated brain extraction inaccurately identified large brain patches within 5 slices outside of the head. This pig exhibited a Dice coefficient of 0.92, which improved to 0.957 after LCC filtration. The LCC filter improved Dice coefficients for all other pigs as well. However, this benefit was smaller due to the lower volumes of outlier voxels; for the remaining 31 pigs, the maximum improvement in Dice coefficient was 0.008 and the mean improvement was 0.001. The final brain extractions, after application of the LCC filter, exhibited Dice coefficients in the range of 0.95–0.97 (mean: 0.961, standard deviation: 0.0036, see histogram in Fig 5(A)), and HD values in the range of 4.1–8.3 voxels (mean: 5.48, standard deviation: 1.16, see histogram in Fig 5(B)), or 2.5–5.0 mm (mean: 3.3, standard deviation: 0.7). The Pearson correlation coefficient, $R$, of the volumes, $V_{manual}$ and $V_{machine}$, of the manual and machine-generated masks was R = 0.90, with p<0.001 (see Fig 6).

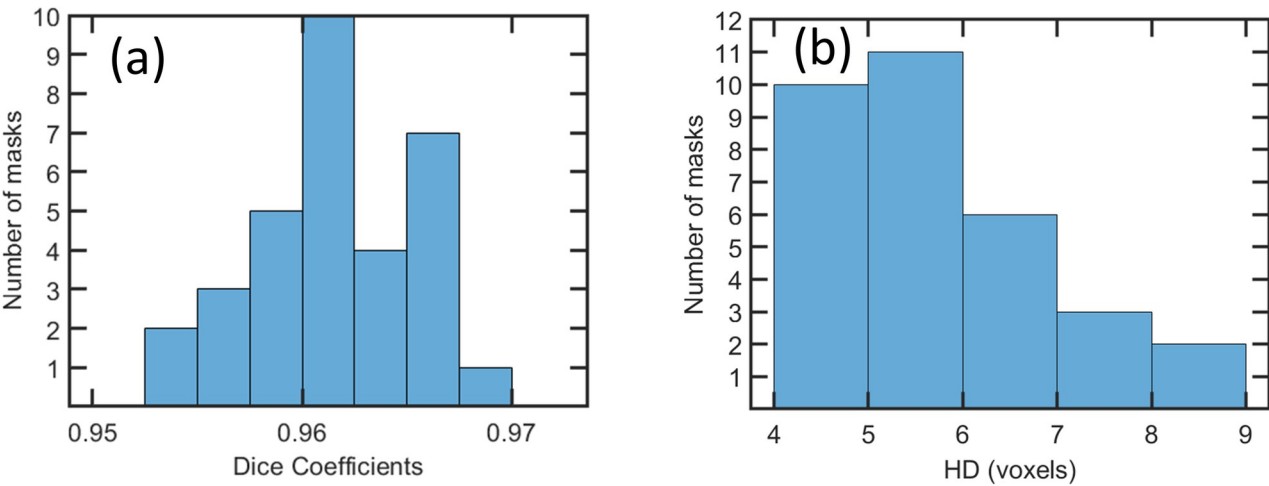

**Fig 5.** Histograms of (a) Dice coefficients and (b) Hausdorff Distance (HD) values calculated from each of the 32 test cases after LCC filtration.

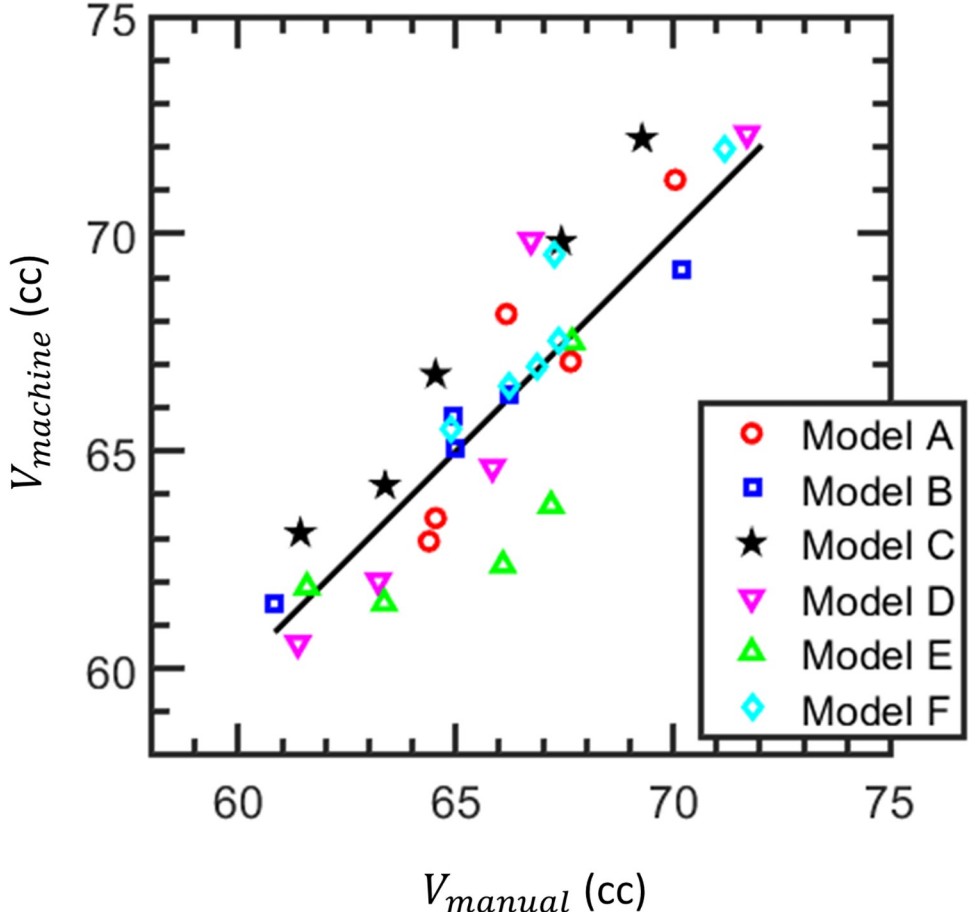

**Fig 6. The consistency of brain volumes, $V_{manual}$ and $V_{machine}$, from manual and machine brain extractions, respectively.** The solid line indicates unity and the colors denote the models used. Brain volume units are cubic centimeters.

## Discussion

We have shown that Mask R-CNN trained on manually generated masks can be used to perform accurate piglet brain extractions. However, for one of the 32 test cases, a model failed to identify brain tissue within several slices. This problem was eliminated when the evaluation was repeated after the removal of an unusually bright region of subcutaneous fat from the images. It is possible that that problem could have been avoided by using more training data, or by performing a bias field correction, or intensity normalization, before training. Image intensity normalization is often important for automated segmentation [21, 22]. We did not perform this step before brain extraction, because in SPM, a bias field correction is typically done simultaneous to tissue segmentation [23]. However, incorporation of a normalization before brain extraction might improve the generalization of our model to other scanning conditions, such as different scanners and coils.

Dice coefficients were >0.95 and HD values were <5 mm; these are similar to values that have been achieved by neural networks for the skull-stripping of non-human primates [11], rodents [24, 25], and piglets [12]. In contrast to our study, Ref [12] employed a 3D patch-based U-net architecture. The potential strength of a 3D patch is that the added depth dimension enhances the local information available to the neural network. However, because of the higher memory demands of using 3D images, the model training and inference steps in Ref. [12] were performed on cubic patches of $32^3$ voxels. A potential disadvantage of this approach is that the segmented patches from the test images must be aggregated and reconciled. By contrast, a potential strength of a 2D approach is the potential to train on complete slices, thereby simplifying post-processing. In Ref. [12] the Dice coefficients from the final method, including post-processing, were in the range of 0.94–0.96 (mean 0.952, standard deviation: 0.0069), slightly lower than the Dice coefficients observed in our study (0.95–0.97, mean: 0.961, standard deviation: 0.0036). Similarly, HD values from Ref. [12] were 5.4–14.3 voxels (mean: 8.51, standard deviation: 2.20), slightly higher than the HD values of our study (4.1–8.3 voxels, mean: 5.48, standard deviation: 1.16). The higher Dice coefficients and lower HD values of our study could be influenced by multiple factors, including image quality differences, which are sensitive to factors such as age differences in piglets, equipment used, and acquisition times. Clearly, a quantitatively accurate comparison of brain extraction methods would require use of the same MRI data. It is possible that the differences in performance metrics were driven by the choice of network architecture. Direct comparisons of Mask R-CNN and U-Net architectures have favored U-Net architectures for segmentation [26] and Mask R-CNN object detection [27, 28]; giving rise to a hybrid methods that exploit the relative strengths of both methods [29], or that improve upon segmentation abilities of Mask R-CNN [30]. Despite the potential drawbacks of Mask R-CNN for segmentation, our results demonstrate that it can be suitable for this application.

Improvements to our approach could be implemented in a variety of ways. Training with a larger sample size is expected to increase performance and accuracy of machine learning algorithms [31]. Performance may also be improved by hyperparameter tuning and optimization [32, 33]. Improved performance might have been obtained by using 3D Mask R-CNN; however we employed 2D Mask R-CNN to obtain shorter training times [34]. The use of 3D, or 2.5D, segmentation has the potential to create a brain mask with greater smoothness at the edge of the brain between adjacent slices. However, the non-smoothness of the automated segmentations is similar to that seen with manual brain extractions, which are also performed in 2D. Also, any inaccuracies due to non-smoothness appear to be localized in the CSF layer surrounding the brain and unlikely to influence the results of automated segmentation of grey matter on the edge of the brain, performed on the extracted brain images. Brain exaction

performance may also be improved by using quantitative imaging techniques, such as MT saturation [35, 36], to improve the contrast between brain and non-brain tissues. Further research is required to access whether network architectures such as U-Net [37], may improve upon results obtained with Mask R-CNN.

In summary, the use of automated brain extraction has the potential to reduce analysis time because it requires minimal supervision. This process is scalable to a high number of piglets, avoiding complications and inconsistencies that might arise from having multiple raters perform manual brain extractions. The effectiveness of Mask R-CNN for performing piglet brain extractions implies that it may be a useful tool for segmenting sub-regions of the brain. Further research is needed to assess whether such an approach may compliment or improve upon existing methods for volumetric analysis of piglet brain MRI data [4, 23, 38].

## Acknowledgments

This work was conducted in part at the Biomedical Imaging Center of the Beckman Institute for Advanced Science and Technology at the University of Illinois Urbana-Champaign (UIUC-BI-BIC).

## Author Contributions

**Conceptualization:** Kayla L. Stanke, Ryan J. Larsen, Brian J. Leyshon.

**Data curation:** Kayla L. Stanke, Ryan J. Larsen.

**Formal analysis:** Kayla L. Stanke, Ryan J. Larsen.

**Funding acquisition:** Ryan J. Larsen, Laurie Rund, Brian J. Leyshon, Andrew J. Steelman.

**Investigation:** Kayla L. Stanke, Ryan J. Larsen, Laurie Rund, Brian J. Leyshon, Allison Y. Louie.

**Methodology:** Kayla L. Stanke, Ryan J. Larsen.

**Project administration:** Ryan J. Larsen, Laurie Rund, Brian J. Leyshon.

**Resources:** Ryan J. Larsen, Laurie Rund, Allison Y. Louie.

**Software:** Kayla L. Stanke, Ryan J. Larsen.

**Supervision:** Ryan J. Larsen, Laurie Rund, Brian J. Leyshon, Andrew J. Steelman.

**Validation:** Kayla L. Stanke, Ryan J. Larsen.

**Visualization:** Kayla L. Stanke, Ryan J. Larsen.

**Writing – original draft:** Kayla L. Stanke, Ryan J. Larsen.

**Writing – review & editing:** Kayla L. Stanke, Ryan J. Larsen, Brian J. Leyshon, Andrew J. Steelman.

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
