## [Decision Letter · Decision Letter 0]

16 Jul 2022

PONE-D-21-37916Automated identification of piglet brain tissue from MRI images using Region-Based Convolutional Neural NetworksPLOS ONE

Dear Dr. Larsen,

Thank you for submitting your manuscript to PLOS ONE. After careful consideration, we feel that it has merit but does not fully meet PLOS ONE’s publication criteria as it currently stands. Therefore, we invite you to submit a revised version of the manuscript that addresses the points raised during the review process.

We look forward to receiving your revised manuscript.

Kind regards,

Zhishun Wang, Ph.D.

Academic Editor

PLOS ONE

Journal Requirements:

“The study was funded by Abbott Nutrition, www.abbott.com, grant number 00490326 to A.S. The funder aided with study design, data analysis, decision to publish, and preparation of the manuscript. We thank the staff members of the Biomedical Imaging Center at the Beckman Institute for Advanced Science and Technology for imaging support.  “

 “The study was funded by Abbott Nutrition, www.abbott.com, grant number 00490326 to

A.S. The funder aided with study design, data analysis, decision to publish, and

preparation of the manuscript.”

Reviewers' comments:

Reviewer's Responses to Questions

**Comments to the Author**

1. Is the manuscript technically sound, and do the data support the conclusions?

Reviewer #1: Yes

Reviewer #2: Partly

2. Has the statistical analysis been performed appropriately and rigorously? 

Reviewer #1: N/A

Reviewer #2: No

3. Have the authors made all data underlying the findings in their manuscript fully available?

Reviewer #1: Yes

Reviewer #2: No

4. Is the manuscript presented in an intelligible fashion and written in standard English?

Reviewer #1: Yes

Reviewer #2: No

5. Review Comments to the Author

Reviewer #1: Overall, the segmentation results appear to be very good, and the cross-fold validation study is rigorous. Splitting the folds over pigs (subject-wise split) avoids data contamination issues between training and testing, which is good.

However, I do not like how the outlier case of one pig was manually-corrected and then the segmentation values were used to report final Dice scores. Manual-correction is fine, but I feel that this is a biased presentation of results. Instead, I encourage you to report both the Dice overlap scores using the original data (that includes the failure case) as well as the results after manual correction for full transparency of results. Imaging artefacts happen all the time, and I think presenting these full results would be more useful for readers.

What form of image intensity normalization was performed? Intensity normalization for MR images is often a critical component. It might not be problem here since all images were from the same scanner, but it would be problem if different scanners were used (or the same scanner was used after any upgrades). Intensity normalization could also help to control for intensity outliers (as seen in your failure case). Please elaborate on any normalization methods used.

Largest connected component filtering was performed as a post-processing step. It would be helpful to report the Dice segmentation values prior to post-processing as well. This would give readers a sense of how well the original segmentation performed compared to how much the post-processing helped.

The inclusion of Dice overlap as similarity metric is good and gives a measure of overall segmentation performance. But, typically, Dice by itself is not the only metric used for segmentation evaluation. Other metrics that are commonly used are Harsdorff Distance (HD) and Mean Surface Distance. In particular, HD is of interest because this gives a measure of worst-case segmentation performance. Worst case segmentation performance is often of interest to readers. Please consider including HD (or its less extreme variant the 95-th percentile of HD).

Training and inference used 2D image slices. During inference, all 2D slices of a piglet were combined into a 3D volume. This has the potential to result in mask results that are not smooth at the boundary because each 2D slice is independent of its neighbors (this is where 2.5D or 3D processing would help). Please discuss the spatial consistency of results between neighboring slices within the 3D volume.

Grammar/Typographical:

Line 134: “all by the” -> “all but the”

Reviewer #2: I have the following concerns about this manuscript:

1- This manuscript is written by a way different from usual known articles including sectioning and subsectioning, references ..etc.

2- There is no mathematical or statistical analysis at all.

3- The methodology used in training, validation, and testing the used model is unclear.

4- The author should compare their results with related published references performing the segmentation task using the same data set without restriction themselves to the piglet MRI images.

6. PLOS authors have the option to publish the peer review history of their article (what does this mean?). If published, this will include your full peer review and any attached files.

Reviewer #1: No

Reviewer #2: **Yes: **Ashraf A. M. Khalaf

---

## [Author Response · Author response to Decision Letter 0]

12 Dec 2022

Reviewer #1: Overall, the segmentation results appear to be very good, and the cross-fold validation study is rigorous. Splitting the folds over pigs (subject-wise split) avoids data contamination issues between training and testing, which is good.

However, I do not like how the outlier case of one pig was manually-corrected and then the segmentation values were used to report final Dice scores. Manual-correction is fine, but I feel that this is a biased presentation of results. Instead, I encourage you to report both the Dice overlap scores using the original data (that includes the failure case) as well as the results after manual correction for full transparency of results. Imaging artefacts happen all the time, and I think presenting these full results would be more useful for readers.

Response: We thank the reviewer for their helpful review. We agree with the reviewer and we have revised the manuscript to report Dices scores before and after this manual correction.

What form of image intensity normalization was performed? Intensity normalization for MR images is often a critical component. It might not be problem here since all images were from the same scanner, but it would be problem if different scanners were used (or the same scanner was used after any upgrades). Intensity normalization could also help to control for intensity outliers (as seen in your failure case). Please elaborate on any normalization methods used.

Response: We agree with the reviewer on the importance of intensity. We have revised the manuscript to state that we did not perform normalization, and to state that might have prevented the segmentation error mentioned by the reviewer, and that it would likely be important for use with data from other scanning conditions.

Largest connected component filtering was performed as a post-processing step. It would be helpful to report the Dice segmentation values prior to post-processing as well. This would give readers a sense of how well the original segmentation performed compared to how much the post-processing helped.

Response: We have updated the manuscript to include this information, and to quantify the improvements in Dice coefficients due to largest connected component filtering

The inclusion of Dice overlap as similarity metric is good and gives a measure of overall segmentation performance. But, typically, Dice by itself is not the only metric used for segmentation evaluation. Other metrics that are commonly used are Harsdorff Distance (HD) and Mean Surface Distance. In particular, HD is of interest because this gives a measure of worst-case segmentation performance. Worst case segmentation performance is often of interest to readers. 

Response: We have updated the manuscript to report Hausdorff Distance measures. 

Training and inference used 2D image slices. During inference, all 2D slices of a piglet were combined into a 3D volume. This has the potential to result in mask results that are not smooth at the boundary because each 2D slice is independent of its neighbors (this is where 2.5D or 3D processing would help). Please discuss the spatial consistency of results between neighboring slices within the 3D volume.

Response: We have updated the manuscript with a discussion of the potential advantages of 3D processing for improvement of consistency between slices.

Grammar/Typographical:

Line 134: “all by the” -> “all but the”

Response: Thank you for pointing this out; it has been corrected.

Reviewer #2: I have the following concerns about this manuscript:

1- This manuscript is written by a way different from usual known articles including sectioning and subsectioning, references ..etc.

Response: We thank the reviewer for their review and comments. We believe that the sectioning is consistent with the usual pattern, including Introduction, Methods, Results, and Discussion. The Methods section is sub-sectioned into Animals and Care Practices, MRI Acquisition, Manual Brain Extraction, Automated Brain Extraction, and Validation.

2- There is no mathematical or statistical analysis at all.

Response: We have updated the manuscript to include Hausdorff Distance values, and some descriptive statistics of them. 

3- The methodology used in training, validation, and testing the used model is unclear.

Response: We tested the model using nested cross-validation. We do not have a separate validation set because we did not perform hyperparameter tuning. The method of nested cross-validation and the potential usefulness of hyperparameter tuning is described in the paper. 

4- The author should compare their results with related published references performing the segmentation task using the same data set without restriction themselves to the piglet MRI images.

Response: We agree that the inclusion of more data could strengthen the manuscript; however, we believe that the inclusion of non-piglet MRI images is beyond the scope of the manuscript.

---

## [Decision Letter · Decision Letter 1]

26 Jan 2023

PONE-D-21-37916R1Automated identification of piglet brain tissue from MRI images using Region-based Convolutional Neural NetworksPLOS ONE

Dear Dr. Larsen,

Thank you for submitting your manuscript to PLOS ONE. After careful consideration, we feel that it has merit but does not fully meet PLOS ONE’s publication criteria as it currently stands. Therefore, we invite you to submit a revised version of the manuscript that addresses the points raised during the review process.

We look forward to receiving your revised manuscript.

Kind regards,

Zhishun Wang, Ph.D.

Academic Editor

PLOS ONE

Journal Requirements:

Reviewers' comments:

Reviewer's Responses to Questions

**Comments to the Author**

1. If the authors have adequately addressed your comments raised in a previous round of review and you feel that this manuscript is now acceptable for publication, you may indicate that here to bypass the “Comments to the Author” section, enter your conflict of interest statement in the “Confidential to Editor” section, and submit your "Accept" recommendation.

Reviewer #1: All comments have been addressed

Reviewer #2: (No Response)

2. Is the manuscript technically sound, and do the data support the conclusions?

Reviewer #1: Partly

Reviewer #2: Partly

3. Has the statistical analysis been performed appropriately and rigorously? 

Reviewer #1: N/A

Reviewer #2: No

4. Have the authors made all data underlying the findings in their manuscript fully available?

Reviewer #1: Yes

Reviewer #2: Yes

5. Is the manuscript presented in an intelligible fashion and written in standard English?

Reviewer #1: Yes

Reviewer #2: No

6. Review Comments to the Author

Reviewer #1: The authors have addressed my comments.

The main limitation of the paper’s evaluation is that the segmentation approach is not compared to alternative segmentation methods as a benchmark. For example, comparison to the 3D U-net approach in [12] using the data from this paper would be ideal. The method in [12] is another segmentation approach applied to very similar data, piglet brain MRI. The proposed approach to use Mask R-CNN would be rigorously justified with a direct head-to-head comparison with the 3D U-net using your dataset.

Absent this numerical comparison, additional discussion comparing to the U-net approach to the proposed approach would be helpful in the Discussion section. The performance of [12] is briefly mentioned in the Discussion (line 223), but a more thorough comparison to the proposed method (both strengths and weaknesses) may be warranted since it is a such a similar application area. However, this discussion must note that a true comparison cannot be made because the two approaches used different datasets.

Reviewer #2: It is the second time to review this manuscript, and almost all my concerns didn't addressed:

1- This manuscript is written by a way different from usual known articles including

sectioning and subsectioning, references ..etc.

Response: We thank the reviewer for their review and comments. We believe that the

sectioning is consistent with the usual pattern, including Introduction, Methods,

Results, and Discussion. The Methods section is sub-sectioned into Animals and Care

Practices, MRI Acquisition, Manual Brain Extraction, Automated Brain Extraction, and

Validation.

Reviewer again: Where is the "Conclusion" section?

Please, see one of the papers published y PloS One:

Zhai Y, Davenport B, Schuetz K, Pappu HR (2022) An on-site adaptable test for rapid and sensitive detection of Potato mop-top virus, a soilborne virus of potato (Solanum tuberosum). PLoS ONE 17(8): e0270918.

https://doi.org/10.1371/journal.pone.0270918

2- There is no mathematical or statistical analysis at all.

Response: We have updated the manuscript to include Hausdorff Distance values, and

some descriptive statistics of them.

Reviewer again: Not addressed

The authors can say that they didn't have any mathematical analysis since they are doing empirical research, then the editorial staff take their decision.

3- The methodology used in training, validation, and testing used in the model is unclear.

Response: We tested the model using nested cross-validation. We do not have a

separate validation set because we did not perform hyperparameter tuning. The

method of nested cross-validation and the potential usefulness of hyperparameter

tuning is described in the paper.

Reviewer again: point Addressed

4- The authors should compare their results with related published references

performing the segmentation task using the same data set without restriction

themselves to the piglet MRI images.

Reviewer again: Not addressed

My opinion is: The comparison with the state-of-the-art is necessary or at least the authors should describe why their approach gives sufficient results that will e an added value to this research area without evidences!. Especially the tool used in this research (Mask R-CNN) is not new!

-------------------

New sample comments:

A- Line 17, the first sentence in the "Abstract" : Magnetic Resonance Imaging.... >>> Magnetic Resonance Imaging (MRI)....

B- How can we evaluate the automated technique visually?

Line 158 : "Visual inspection of the brain extractions reveals good accuracy of automatic brain extractions

(Fig 3)....

7. PLOS authors have the option to publish the peer review history of their article (what does this mean?). If published, this will include your full peer review and any attached files.

Reviewer #1: No

Reviewer #2: No

---

## [Author Response · Author response to Decision Letter 1]

21 Mar 2023

We have written our responses to the reviewers in the attached file, "Response to Reviewers".

---

## [Decision Letter · Decision Letter 2]

13 Apr 2023

Automated identification of piglet brain tissue from MRI images using Region-based Convolutional Neural Networks

PONE-D-21-37916R2

Dear Dr. Larsen,

We’re pleased to inform you that your manuscript has been judged scientifically suitable for publication and will be formally accepted for publication once it meets all outstanding technical requirements.

Kind regards,

Zhishun Wang, Ph.D.

Academic Editor

PLOS ONE

Additional Editor Comments (optional):

Reviewers' comments:

Reviewer's Responses to Questions

**Comments to the Author**

1. If the authors have adequately addressed your comments raised in a previous round of review and you feel that this manuscript is now acceptable for publication, you may indicate that here to bypass the “Comments to the Author” section, enter your conflict of interest statement in the “Confidential to Editor” section, and submit your "Accept" recommendation.

Reviewer #1: All comments have been addressed

Reviewer #2: All comments have been addressed

2. Is the manuscript technically sound, and do the data support the conclusions?

Reviewer #1: Yes

Reviewer #2: Partly

3. Has the statistical analysis been performed appropriately and rigorously? 

Reviewer #1: N/A

Reviewer #2: No

4. Have the authors made all data underlying the findings in their manuscript fully available?

Reviewer #1: Yes

Reviewer #2: Yes

5. Is the manuscript presented in an intelligible fashion and written in standard English?

Reviewer #1: Yes

Reviewer #2: Yes

6. Review Comments to the Author

Reviewer #1: While the authors have addressed my comments for the most part, I am a little disappointed that the effort was not made to train another network for comparison. Training and testing another network for benchmarking is standard practice in image analysis and it does not require tremendous resources in the case of a U-net. The discussion comparing to the alternative U-net is ok, but it is a lesser replacement for head-to-head comparison.

Reviewer #2: Almost all my previous comments in the previous review round have been addressed in this revised version.

7. PLOS authors have the option to publish the peer review history of their article (what does this mean?). If published, this will include your full peer review and any attached files.

Reviewer #1: No

Reviewer #2: **Yes: **Ashraf A. M. Khalaf

---

## [Editor Report · Acceptance letter]

19 Apr 2023

PONE-D-21-37916R2 

Automated identification of piglet brain tissue from MRI images using Region-based Convolutional Neural Networks 

Dear Dr. Larsen:

I'm pleased to inform you that your manuscript has been deemed suitable for publication in PLOS ONE. Congratulations! Your manuscript is now with our production department. 

Kind regards, 

on behalf of

Dr. Zhishun Wang 

Academic Editor

PLOS ONE